# Tropospheric and stratospheric NO retrieved from ground-based FTIR measurements

Minqiang Zhou[1], Bavo Langerock[1], Corinne Vigouroux[1], Bart Dils[1], Christian Hermans[1], Nicolas Kumps[1], Jean-Marc Metzger[3], Emmanuel Mahieu[4], Pucai Wang[2,5,6], and Martine De Mazière[1]

[1]Royal Belgian Institute for Space Aeronomy (BIRA-IASB), Brussels, Belgium
[2]CNRC & LAGEO, Institute of Atmospheric Physics, Chinese Academy of Sciences, Beijing, China
[3]UMS 3365 – OSU Réunion, Université de La Réunion, Saint-Denis, Réunion, France
[4]Institut d'Astrophysique et de Géophysique, UR SPHERES, Université de Liège, Liège, Belgium
[5]University of Chinese Academy of Sciences, Beijing, China
[6]Xianghe Observatory of Whole Atmosphere, Institute of Atmospheric Physics, Chinese Academy of Sciences, Xianghe, China

**Correspondence:** Minqiang Zhou (minqiang.zhou@aeronomie.be)

**Abstract.** Nitric oxide (NO) is a key active trace gas in the atmosphere, which contributes to form harmful ozone in the troposphere and to the destruction of ozone in the stratosphere. In this study, we present the NO retrieval from ground-based Fourier-transform infrared (FTIR) solar absorption spectrometry measurements at a polluted site (Xianghe, China) and a background site (Maïdo, Reunion Island). The Degree Of Freedom (DOF) of the NO retrieval is 2.3±0.4 ($1\sigma$) at Xianghe and 1.3±0.1 at Maïdo.

By looking at the FTIR NO retrievals at Xianghe and Maïdo, we find that the stratospheric NO partial column is large in summer as compared to winter at both sites, and the seasonal variation of the FTIR stratospheric NO partial columns is consistent with that observed by the co-located Michelson Interferometer for Passive Atmospheric Sounding (MIPAS) satellite measurements. A large diurnal variation in the stratospheric NO partial column is observed by the FTIR measurements at Maïdo, with an increase from the early morning to about 14:00 local time and a decrease thereafter.

Due to the low NO concentration near the surface, the FTIR NO retrieval is only sensitive to the stratosphere at Maïdo. The high NO mole fraction near the surface at Xianghe allows us to derive tropospheric and stratospheric NO partial columns separately, albeit the tropospheric column is very difficult to retrieve in summer (June-August) because of the high water vapor abundance. A good correlation is found between the NO observed by the FTIR measurements and other air pollutants ($NO_2$ and CO) in the troposphere at Xianghe. It is the first study of a successful analysis of NO in the troposphere from a ground-based FTIR site. The tropospheric and stratospheric NO retrieval might be possible at other potential FTIR sites inside/near large cities with enhanced levels of NO near the surface.

## 1 Introduction

Nitric oxide (NO) is a major component of the nitrogen oxides family ($NO_x = NO + NO_2$), which plays key roles in atmospheric chemistry. In the troposphere, NO is an air pollutant, related to the formation of ground-level ozone ($O_3$), peroxyacetyl nitrate

(PAN), nitric acid (HNO$_3$) and aerosols (Crutzen, 1979; Ng et al., 2007; Monks et al., 2015). NO sources near the surface are mainly of anthropogenic origin. Delmas et al. (1997) pointed out that about 50% of the NO emissions are caused by the combustion of fossil fuel, and about 20% are from the biomass burning. The remaining 30% are mainly from natural lightning and microbial activity in soils. In the stratosphere, NO participates in an important set of catalytic reactions which deplete

ozone (Crutzen, 1970). Stratospheric NO is mainly coming from the oxidation of nitrous oxide (N$_2$O), which is a stable trace gas that can be transported upward to the stratosphere in the tropical region. As atmospheric N$_2$O has been increasing since the 1970s mainly due to increasing use of fertilizers (Park et al., 2012), stratospheric ozone depletion caused by NO$_x$ will play a more important role in the future, especially as the stratospheric chlorine burden is declining (Portmann et al., 2012). In the mesosphere and thermosphere, NO is formed by energetic particle precipitation (Randall et al., 2007), which can be

transported downward to the stratosphere affecting the ozone chemistry, especially in the winter polar region (Meraner and Schmidt, 2016).

Space-borne sensors, e.g. the Atmospheric Chemistry Experiment - Fourier Transform Spectrometer (ACE-FTS) making solar occultation measurements (Bernath et al., 2005) and the Michelson Interferometer for Passive Atmospheric Sounding (MIPAS) looking at thermal emission at the limb (Fischer et al., 2008), can provide global NO distributions. However, these

satellite measurements provide almost no information in the troposphere. Moreover, it is difficult to derive the diurnal variation of NO from these satellite measurements. Because of the weak intensities of the NO absorption lines (Gordon et al., 2017), as far as we know, there is no nadir-looking satellite to measure the NO near the surface. Ground-based Fourier-transform infrared (FTIR) spectrometers affiliated with the Network for Detection of Atmospheric Composition Change (NDACC) (De Mazière et al., 2018) record direct solar absorption spectra in the infrared region with a high spectral resolution (0.0035 - 0.005 cm$^{-1}$)

under clear sky conditions. More than 20 atmospheric species can be retrieved from the FTIR observed spectra, and the data have been widely used to investigate the change of atmospheric composition, to support satellite validation and model verification (De Mazière et al., 2018). However, until now, there are few studies focusing on FTIR NO retrieval. Notholt et al. (1995) showed that the NO total columns can be retrieved from the FTIR spectra with a high spectral resolution of 0.0035 cm$^{-1}$ at Ny-Alesund. Wiacek et al. (2006) succeeded in retrieving NO in the stratosphere, mesosphere and in the lower thermosphere

from the ground-based FTIR measurements at Toronto and Eureka, but they found that there is almost no information in the troposphere for the FTIR NO retrievals at Eureka, and even for measurements taken in the Toronto mega-city.

In this study, we investigate NO retrievals from ground-based FTIR measurements with a focus on the retrieved profile in the troposphere and stratosphere at two different sites: Xianghe, a polluted site in China and the Maïdo observatory on Reunion Island, a background site. The aims of this study are 1) to investigate whether it is possible to retrieve NO partial columns

in the troposphere and stratosphere from the ground-based FTIR measurements, especially at the polluted site Xianghe; 2) to understand the diurnal, synoptic and/or seasonal variations of NO partial columns observed by the ground-based FTIR measurements in the stratosphere (and troposphere if possible) at Xianghe and Maïdo, together with other measurements, such as co-located satellite measurements. In Section 2, we give a brief introduction to the sites and the FTIR measurement technique, and discuss the FTIR NO retrieval strategy and retrieval uncertainties. In Section 3, we discuss the time series of the

FTIR NO retrievals at Xianghe and Maïdo, including the diurnal and seasonal variations of the FTIR retrieved partial columns

of NO in the stratosphere. In addition, the FTIR NO retrieved stratospheric partial columns are compared with the co-located MIPAS satellite measurements. Moreover, the FTIR retrieved tropospheric partial columns of NO at Xianghe are discussed in Section 4. Finally, conclusions are summarized in Section 5.

## 2 Measurement sites and retrieval strategy

### 2.1 FTIR sites

- Xianghe (39.75 °N, 116.96 °E; 50 m a.s.l.) is located in a polluted urban region in North China. It is about 70 km southeast of Beijing. A Bruker IFS 125HR spectrometer was installed at Xianghe and started measuring infrared solar absorption spectra in June 2018 (Zhou et al., 2020). The Xianghe site is operated in the Total Column Carbon Observing Network (TCCON) mode by recording near infrared spectra from 4000 to 12000 $cm^{-1}$ using an InGaAs detector (Yang et al., 2020), but infrared spectra with a spectral range from 1800 to 5200 $cm^{-1}$ are also recorded with a liquid $N_2$ InSb cooled detector. The spectra at Xianghe used for the NO retrieval are operated with the NDACC IRWG optical filter no.5 (Blumenstock et al., 2021). The maximum optical path difference (MOPD) is 257 cm, which corresponds to a spectral resolution of 0.0035 $cm^{-1}$. There are normally 1-5 observed spectra on each measurement day. According to the Emissions Database for Global Atmospheric Research (EDGAR v4.3.2) (Crippa et al., 2018), the $NO_x$ annual emission at Xianghe in 2012 is larger than 1000 tonnes/yr/(0.1 deg)$^2$, which is one of the largest $NO_x$ emission rates around the world. The NO concentration near the surface in this region was reported of about 5-20 ppb (Tang et al., 2009). The high NO concentration provides an opportunity to study whether it is possible to retrieve tropospheric NO columns from the ground-based FTIR spectra.

- The Maïdo (21.08 °S, 55.38 °E; 2155 m a.s.l.) observatory, located on a mountain at Reunion Island, is about 700 km east of Madagascar in the southern hemisphere tropical region. The Bruker IFS 125HR spectrometer at Maïdo is affiliated with the NDACC-InfraRed Working Group (IRWG) (De Mazière et al., 2018), and has been measuring solar absorption spectra quasi continuously since March 2013 (Zhou et al., 2016). The infrared spectra in the spectral range from 800 to 2000 $cm^{-1}$ are recorded with a liquid $N_2$ cooled MCT detector and the infrared spectra in the spectral range from 2000 to 5200 $cm^{-1}$ are recorded with a liquid $N_2$ cooled InSb detector. The spectra at Maïdo used for the NO retrieval are also operated with the NDACC IRWG optical filter no.5. Different from Xianghe, the spectra at Maïdo used for the NO retrieval are operated with 2 MOPD of 120 cm and 257 cm, corresponding to spectral resolutions of 0.0072 $cm^{-1}$ and 0.0035 $cm^{-1}$, respectively. The short MOPD is operated with a large SZA (>60°) to reduce the uncertainty of the light path change. The long MOPD is operated with a small SZA. There are normally 1-10 spectra on each measurement day. Since NO is not among the baseline species in the NDACC-IRWG network, it is the first time that we study the NO retrieval at the Maïdo site. As the $NO_x$ surface concentration is low at Maïdo, with a typical value of 0.1-0.5 ppb (Rocco et al., 2020), it can be considered as a background site as compared to Xianghe.

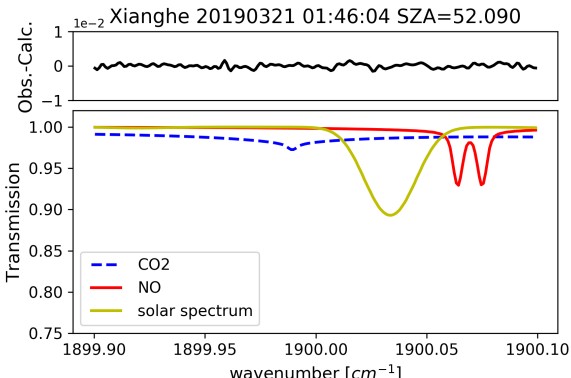

**Figure 1.** An example of the transmittances from NO, carbon dioxide ($CO_2$) and solar lines (lower) and the difference between the observed and calculated spectra (upper) in the NO retrieval window (1899.9-1900.1 cm$^{-1}$) at Xianghe. The measurement time (UTC) together with the solar zenith angle is shown in the title.

## 2.2 FTIR retrieval strategy

The SFIT4 v0.9.4.4 retrieval code, updated from SFIT2 (Pougatchev et al., 1995), based on the optimal estimation method (Rodgers, 2000) is applied to retrieve the NO profile from the infrared spectra observed at Xianghe and Maïdo. A line-by-line model has been implemented in the forward model of the SFIT4 code to calculate the transmittance at a given wavenumber
range:

$$\boldsymbol{Y} = \boldsymbol{F}(\boldsymbol{x}, \boldsymbol{b}) + \varepsilon, \tag{1}$$

where $\boldsymbol{Y}$ is the observed spectrum, $\boldsymbol{F}(\boldsymbol{x}, \boldsymbol{b})$ is the forward model, with the inputs from the retrieved parameters ($\boldsymbol{x}$) and non-retrieved model parameters ($\boldsymbol{b}$) and $\varepsilon$ is the uncertainty. The pressure and temperature dependences of the line shape allow us to retrieve some pieces of vertical information of the target gas. The HITRAN2016 spectroscopy (Gordon et al., 2017) is used
here, and the strongest NO absorption lines are distributed in the range between 1820 and 1930 cm$^{-1}$. In order to select the strong NO lines and to reduce the interference from other species, especially $H_2O$, several windows have been tested. We find that the NO absorption lines at 1900 cm$^{-1}$ are the best choice for ground-based FTIR NO retrieval at Xianghe and Maïdo, which have also been used in the previous studies (Notholt et al., 1995; Wiacek et al., 2006). Figure 1 shows an example of the spectral fitting in the retrieval window at Xianghe. In order to reduce the influence of the interfering species, the column of
$CO_2$ is retrieved simultaneously together with the profile retrieval of NO.

A cost function ($\boldsymbol{J}(\boldsymbol{x})$) is created and the STIT4 algorithm uses an iterative Levenberg-Marquardt method to look for the optimal $\boldsymbol{x}$ to minimize the $\boldsymbol{J}(\boldsymbol{x})$.

$$\boldsymbol{J}(\boldsymbol{x}) = [\boldsymbol{y} - \boldsymbol{F}(\boldsymbol{x})]^T \mathbf{S}_{\epsilon}^{-1} [\boldsymbol{y} - \boldsymbol{F}(\boldsymbol{x})] + [\boldsymbol{x} - \boldsymbol{x_a}]^T \mathbf{S}_a^{-1} [\boldsymbol{x} - \boldsymbol{x_a}], \tag{2}$$

where $\mathbf{S}_a$ is the a priori covariance matrix, and $\mathbf{S}_\epsilon$ is the measurement covariance matrix. The retrieved state vector ($\boldsymbol{x}_r$) can be written as

$$\boldsymbol{x}_r = \boldsymbol{x}_a + \mathbf{A}(\boldsymbol{x}_t - \boldsymbol{x}_a) + \boldsymbol{\varepsilon}, \tag{3}$$

$$\mathbf{A} = (\mathbf{K}^T \mathbf{S}_\epsilon^{-1} \mathbf{K} + \mathbf{S}_a^{-1})^{-1} \mathbf{K}^T \mathbf{S}_\epsilon^{-1} \mathbf{K}, \tag{4}$$

where $\boldsymbol{x}_a$ and $\boldsymbol{x}_t$ are the a priori and true state vectors (retrieved parameters), respectively, $\boldsymbol{\varepsilon}$ is the total error on the retrieved profile minus the smoothing error, $\mathbf{K}$ is the Jacobian matrix and $\mathbf{A}$ is the averaging kernel (AVK), indicating the sensitivity of the retrieved parameters to the true parameters. A fixed a priori NO profile is used for all the retrievals at one site. The a priori profile of NO at Maïdo is derived from the average of the Whole Atmosphere Community Climate Model (WACCM) monthly means within 1980-2020 (Marsh et al., 2013), which is often used to create the a priori profile within the NDACC-IRWG

community. Note that the daytime and night-time model data are both used in this case. There is an underestimation of NO near the surface from the WACCM model at Xianghe. The NO mole fraction is about 0.2 ppb in the WACCM model while the surface observations in Beijing indicate that NO mole fraction in this region is about 5-20 ppb near the surface during daytime (Tang et al., 2009). Therefore, we use the annual mean in 2018 from the Community Atmosphere Model with Chemistry (CAM-Chem) model (Lamarque et al., 2012) monthly means as the a priori NO profile at Xianghe, with the NO mole fraction

of 9.2 ppb at the surface. Since the top pressure level in the CAM-Chem model is about 1.8 hPa ($\sim$ 50km), the a priori profile of NO at Xianghe above 50 km is taken from the WACCM model.

     The regularization matrix for the NO retrieval is created with the Tikhonov $\mathbf{L}_1$ method (Tikhonov, 1963). $\mathbf{R} = \mathbf{S}_a^{-1} = \alpha \mathbf{L_1}^T \mathbf{L_1}$. To determine the value of $\alpha$, we use the DOF method as the described in Steck (2002). First, we create the a priori covariance matrix ($\mathbf{Sa}$) using the WACCM model monthly means. Second, the retrieval is operated using the optimal

estimation method (OEM), and the DOF from the OEM is about 1.3 at Maïdo. Finally, we tune the $\alpha$ value to get a similar DOF using the Tikhonov method, implying $\alpha$ = 50. In this study, we use the same $\alpha$ at both sites. The covariance matrix of the measurement ($\mathbf{S}_\epsilon$) is calculated as $1/\mathrm{SNR}^2$ for the diagonal values and 0 for the off-diagonal values. As a result, the AVK is affected by the signal-to-noise ratio (SNR) of the spectrum. Although there are only $CO_2$ and solar lines in our retrieval window, there are many strong water vapor lines adjacent to the window. Therefore, the SNR in this region is strongly affected

by the water vapor abundance. The SNR is defined as the ratio of the maximum intensity of the spectra in the NO retrieval window to the root mean square error of the spectra in the noise window between 1650 and 1700 $\mathrm{cm}^{-1}$. Figure 2 shows several typical spectra observed in summer and winter at Xianghe and Maïdo. According to the National Centers for Atmospheric Prediction (NCEP) reanalysis data (Kalnay et al., 1996), the mean total columns of $H_2O$ are 3.8 $\times 10^{22}$ $molecules/cm^2$ (7.5 $\times 10^{22}$ $molecules/cm^2$ in summer and 1.6 $\times 10^{22}$ $molecules/cm^2$ in winter) at Xianghe and 2.6 $\times 10^{22}$ $molecules/cm^2$

(3.8 $\times 10^{22}$ $molecules/cm^2$ in summer and 1.0 $\times 10^{22}$ $molecules/cm^2$ in winter) at Maïdo. The $H_2O$ interference is more important at Xianghe as compared to Maïdo. As a result, the SNR of the spectrum is less than 50 in summer and about 500 in winter at Xianghe, and it is about 200 in summer and about 700 in winter at Maïdo. HBr cell measurements are operated at both sites. The instrument line shape (ILS) parameters are retrieved from the cell measurements by the LINEFIT14.5 algorithm (Hase et al., 1999), and the LINEFIT outputs are used as the ILS inputs in the SFIT4 algorithm. The solar line list is included

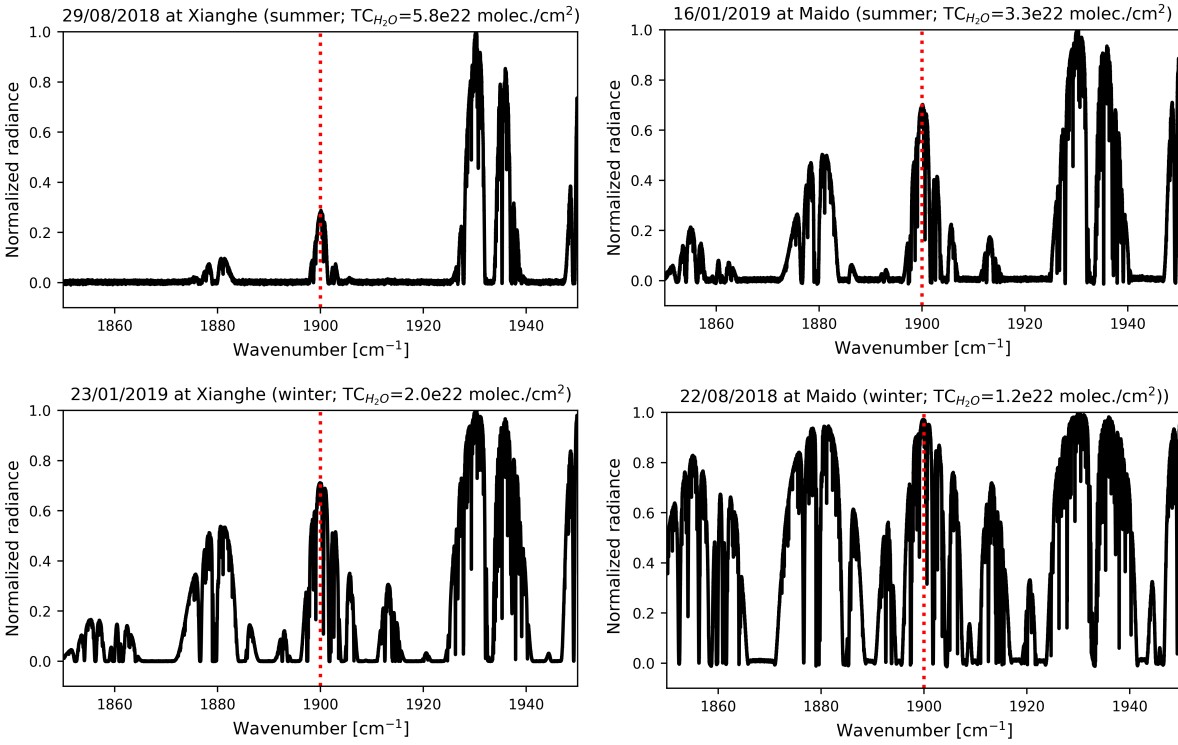

**Figure 2.** The normalized spectra in summer and winter at Xianghe and Maïdo, together with the total column of water vapor on these days. The red dashed line indicates the retrieval window for NO.

in the SFIT4 code, named 120621_solar, provided by Frank Hase (KIT), and the solar line intensity and shift are retrieved simultaneously in the NO retrieval. Table 1 summarizes the NO retrieval strategy used in this study.

**Table 1.** The retrieval strategy of FTIR NO retrieval in this study.

| Parameter | Setting |
|---|---|
| Retrieval window | 1899.90-1900.10 cm$^{-1}$ |
| Profile retrieval | NO |
| Column retrieval | CO$_2$ |
| A priori profile | WACCM (+ CAM-Chem at Xianghe) |
| Spectroscopy | HITRAN2016 |
| Regularization | Tikhonov (alpha = 50) |
| SNR | Calculated from the spectra |
| ILS | LINEFIT retrievals |
| Solar lines | The intensity and shift are retrieved |

Figure 3 shows the a priori and retrieved NO profiles in the vertical range between the surface and 70 km, together with typical AVKs at Xianghe and Maïdo, respectively. At both sites, the maximum value of the NO mole fraction occurs at about 45 km in the stratosphere. The retrieved NO mole fraction is much larger than the a priori in the stratosphere at both sites. The a priori NO profile is created as the average of the model monthly means including both daytime and night-time data, while the NO mole fraction in the stratosphere at night is several orders of magnitude less than that during the daytime (Kondo et al., 1990; Dubé et al., 2020). Therefore, in the stratosphere the FTIR retrievals during the daytime are much larger than the a priori profile. The retrieved NO mole fraction near the surface is about 10 ppb at Xianghe, which is comparable to the observations in Beijing of 5-20 ppb during daytime (Tang et al., 2009). The NO mole fraction near the surface is about 0.01 ppb at Maïdo. The AVK at Xianghe shows that only the layers below 2 km have sensitivity near the surface, while other layers are mainly sensitive to the stratosphere. The AVK at Maïdo shows that the kernels of the troposphere and stratosphere both peak at the stratosphere ($\sim$35 km), indicating that there is almost no information in the lower troposphere. According to the NCEP data, the mean tropopause heights at Xianghe and Maïdo are about 12 km and 16 km, respectively. In this study, we take the vertical range between the surface and the tropopause height (12 km at Xianghe and 16 km at Maïdo) as the troposphere and the vertical range between the tropopause height and 60 km as the stratosphere.

The trace of the AVK matrix is the degree of freedom (DOF) for signal, indicating the number of individual pieces of information. The mean DOFs of the retrieved NO profiles over the entire datasets are 2.3$\pm$0.4 (1$\sigma$) at Xianghe and 1.3$\pm$0.1 at Maïdo, respectively. Figure 4 shows the time series of the DOF in the troposphere, stratosphere, and above 60 km between July 2018 and June 2020 at Xianghe, and between March 2013 and December 2019 at Maïdo. The DOF in the vertical range above 60 km is 0.20$\pm$0.06 at Xianghe and 0.10$\pm$0.02 at Maïdo. The DOF in the stratosphere is generally between 1.0 and 1.8 at both sites. The DOF in the troposphere is 0.78$\pm$0.18 with 90% of DOF from the layers below 2 km at Xianghe, reflecting that the tropospheric NO partial column is actually dominated by the NO partial column in the boundary layer. At Maïdo, the DOF in the troposphere is only 0.03$\pm$0.02. In the remainder of this study, we consider only the measurements with a tropospheric DOF larger than 0.5 (black dashed line in Figure 4) as the information on the tropospheric NO partial column will in this case comes mostly from the retrieval, and not from the a priori. In total, 472 out of 539 retrievals are selected, with almost none in summer (June to August). As seen in Figure 4, there is a large seasonal variation in the DOF. The DOF is determined by the SNR of the spectrum, which is highly related to the $H_2O$ total column (Figure 5). A large $H_2O$ abundance makes the signal in the selected spectral region weak, leading to a low DOF. The correlation coefficient (R) between DOF and $H_2O$ total column is -0.88. Based on the linear fit, the FTIR NO retrievals at Xianghe with DOF in the troposphere larger than 0.5 are generally occurring when the $H_2O$ total column is less than 5.7 $\times10^{22}$ $molecules/cm^2$. In summary, we cannot retrieve NO in the troposphere at Maïdo, because the NO mole fraction near the surface ($NO_{surf}$) is low, with a typical value of less than 0.1 ppb. At Xianghe, the spectra recorded under a wet condition (mainly occur in summer) do not allow us to retrieve the tropospheric NO either. In winter, all the retrievals at Xianghe provide both tropospheric and stratospheric NO partial columns (Figures 4 and 5). The retrieved $NO_{surf}$ in winter ranges from 1.3 to 47.2 ppb, with a mean of 11.4 ppb and an std of 10.7 ppb. For all the 240 retrievals in winter, the mean of the $H_2O$ total column is 2.3 $\times10^{22}$ $molecules/cm^2$, and the mean of the SZA is 65.3°. A relatively lower $NO_{surf}$ at Xianghe can be detected under the condition of a low $H_2O$ total column and a large SZA. For

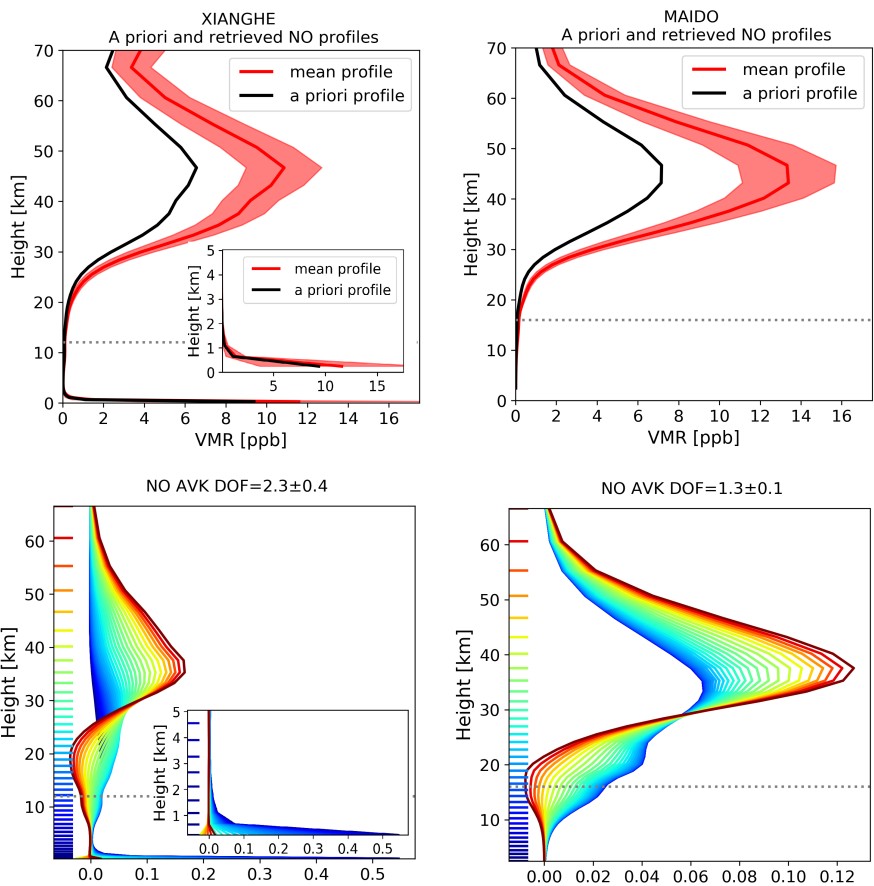

**Figure 3.** The a priori and retrieved NO profiles at Xianghe and Maïdo (upper panels), together with a typical averaging kernel (AVK) at each site (bottom panels). To better visualize the change near the surface, a zoom on the vertical range between 0 and 5 km is also shown for Xianghe. The red shadow in the upper panels is the standard deviation of the retrieved profiles. The dashed line indicates the tropopause height.

example, if we only select the retrievals with the $NO_{surf}$ less than 3 ppb (26 out of 240), the mean of the $H_2O$ total column becomes $1.7 \times 10^{22}$ $molecules/cm^2$, and the mean of the SZA is $68.1°$.

Table 2 lists the systematic and random retrieval uncertainties of the NO total column at Xianghe and Maïdo, including the contributions from the major uncertainty sources, which are estimated based on the optimal estimation method (Rodgers, 2000). It is assumed that the systematic uncertainty of a priori profile is 10%, and the random uncertainty of a priori profile is calculated from the covariance matrix of the monthly a priori profiles in 2018. The measurement error matrix is created by the SNR of the spectra, so that each individual spectrum has a different measurement uncertainty matrix. The temperature systematic and random uncertainties are derived from the mean and standard deviation (std) of the differences between the NCEP and ECMWF reanalysis data. We set the systematic uncertainty of the NO spectroscopy to 10% according to the HITRAN2016

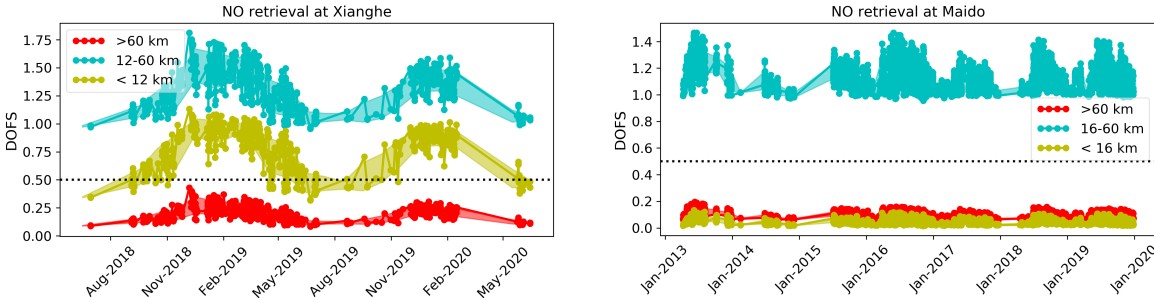

**Figure 4.** The series of the DOF daily mean (dots) and the std (shadow) in the troposphere, stratosphere and above at Xianghe (left) and Maïdo (right). The black dashed line is at DOF of 0.5.

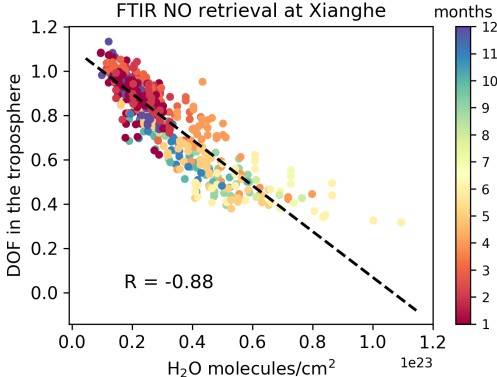

**Figure 5.** The scatter plots between the DOF in the troposphere and the $H_2O$ total columns from the NCEP data, coloured with the measurement month. The black dashed line is the linear fit, and R is the correlation coefficient.

linelist (Gordon et al., 2017), and assume that there is no random uncertainty for the spectroscopy. The systematic and random uncertainties of solar zenith angles (SZA) are set to 0.1% and 0.5%, respectively. The retrieved parameter contains the interfering species ($CO_2$), the solar line intensity and shift, and the slope. The systematic and random uncertainties of $CO_2$ are set to 5 and 10%, respectively. We set 0.1%, 1.0% and 0.5% to both the random and systematic uncertainties of the slope, the solar

5   line intensity and the solar line shift, respectively. In total, the systematic uncertainties of the FTIR NO retrieved total columns are similar at Xianghe and Maïdo (∼10.3%), and dominated by the uncertainty of the spectroscopy. The random uncertainty of retrieved NO total column is estimated to be 13.5% at Xianghe, which is larger than that of 4.2% at Maïdo. The random uncertainty is mainly coming from the smoothing error and the measurement uncertainty, where the large smoothing error at Xianghe is large due to the strong NO variation near the surface, and the large measurement uncertainty at Xianghe is coming

10   from the low SNR of the spectra. At Maïdo, the systematic uncertainties of NO stratospheric partial column and total column are similar. The random uncertainty of NO stratospheric partial column is less than the random uncertainty of NO total column,

mainly due to a smaller smoothing error. At Xianghe, the systematic and random uncertainties of the NO partial columns in the troposphere and stratosphere are also shown in Table 2. The systematic and random uncertainties of the stratospheric NO partial column are 10.2% and 4.4%, respectively. The systematic and random uncertainties of the tropospheric NO partial column are 10.5% and 18.0%, respectively. The random uncertainty of the tropospheric NO partial column is larger as compared to the stratospheric partial column, and it is mainly coming from the smoothing error and the measurement uncertainty.

## 3 Stratospheric NO partial column

### 3.1 Diurnal variation

Due to photochemical reactions (Kondo et al., 1990), a large diurnal variation of the stratospheric NO concentration is excepted. Figure 6 shows that the diurnal variations of stratospheric NO partial columns in all months at Maïdo, together with the SZA of the measurements. The stratospheric NO partial columns are fitted with a second order polynomial fitting ($y(t) = a + bt + ct^2$; $t$ is a fraction of local hour). No fitting is applied at Xianghe due to the lack of measurements, especially before 9:00 and after 16:00. Note that the fitted line is also plotted at hours with no measurements, but it does not represent the physical reality due to the absence of data.

At Maïdo, the measurement often starts at 7:00 and stops at 17:30. The FTIR measurements show that the stratospheric NO partial column increases with time until about 14:00, and it starts decreasing afterwards. The decrease in NO after 14:00 indicates that more NO is converted to nitrogen dioxide ($NO_2$) than the NO formed from $NO_2$ at that time. This type of diurnal variation is consistent throughout the whole year at Maïdo. Based on the second order polynomial fittings, it is found that the maximum stratospheric NO partial column occurs at 14.3±0.4 hours, which is 1.9±0.6 hours after the time of the minimum of the SZA (about 12:20). There is almost no cloud in the stratosphere, so the solar radiation intensity is directly proportional to $cos(SZA)$. The stratospheric NO partial columns increase with time in the morning, and we find that there is a good linear relationship between the stratospheric NO partial column and the solar radiation intensity ($cos(SZA)$) between 6:00 and 12:20 (Figure 7), with the R of 0.80 at Xianghe and 0.74 at Maïdo. The fitted slope at Xianghe is 2.01±0.16×10^{15}, which is close to the slope of 1.87±0.12×10^{15} at Maïdo. The FTIR measurements show that the speed on the formation of stratospheric NO in the morning at Maïdo is similar to that at Xianghe.

### 3.2 Time series and seasonal variation

Figure 8 shows the time series of the FTIR NO retrieved partial columns in the stratosphere at Xianghe and Maïdo. In order to derive the seasonal variation, the daily means $y(t)$ are fitted by a periodic function

$$y(t) = A_0 + A_1 t + \sum_{k=1}^{3} (A_{2k} \cos(2k\pi t) + A_{2k+1} \sin(2k\pi t)), \tag{5}$$

where $t$ is a fraction of year, $A_0$ is the offset, $A_1$ is the long-term trend, and $A_2$ to $A_7$ are the periodic amplitudes, representing the seasonal variation. The annual relative change in unit of % $y^{-1}$ is relative to the mean of data used in the trend analysis.

**Table 2.** The retrieval uncertainties of the NO total and partial columns at Xianghe and Maïdo. All uncertainties are in the unit of percentage unites (%).

| Site | Xianghe | | | | | | Maïdo | | | |
|---|---|---|---|---|---|---|---|---|---|---|
| | Total column | | Troposphere (0-12 km) | | Stratosphere (12-60 km) | | Total column | | Stratosphere (16-60 km) | |
| | Systematic | Random | Systematic | Random | Systematic | Random | Systematic | Random | Systematic | Random |
| Smoothing | 1.5 | 12.0 | 1.7 | 16.2 | 1.5 | 2.4 | 0.2 | 3.6 | 0.1 | 1.5 |
| Measurement | - | 6.0 | - | 7.5 | - | 2.5 | - | 1.7 | - | 1.4 |
| Retrieved parameters | 1.2 | 1.2 | 1.1 | 1.1 | 1.5 | 1.5 | 0.4 | 0.4 | 0.5 | 0.5 |
| Temperature | 1.4 | 0.6 | 1.5 | 1.2 | 2.8 | 1.5 | 2.5 | 0.5 | 2.6 | 0.6 |
| Spectroscopy | 10.1 | - | 10.2 | - | 9.8 | - | 10.0 | - | 9.9 | - |
| Solar zenith angle | 0.3 | 1.5 | 0.4 | 1.8 | 0.3 | 1.6 | 0.2 | 1.0 | 0.2 | 1.0 |
| Total | 10.3 | 13.5 | 10.5 | 18.0 | 10.2 | 4.4 | 10.3 | 4.2 | 10.2 | 2.4 |

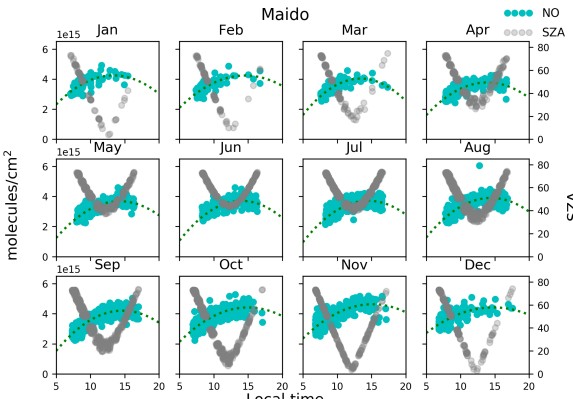

**Figure 6.** The diurnal variation of stratospheric NO partial column at Maïdo, together with the solar zenith angle (grey dots). The stratospheric NO partial columns are fitted with a second order polynomial fitting (cyan dotted line).

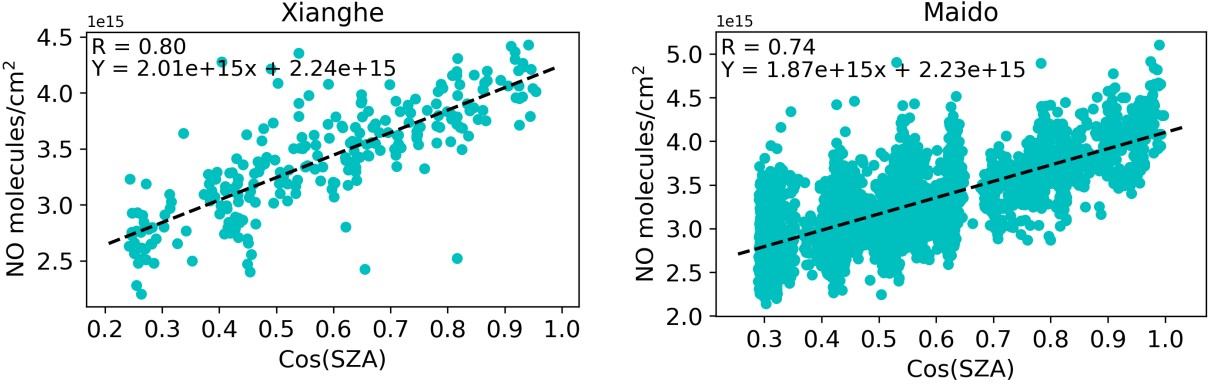

**Figure 7.** The correlation between the stratospheric NO partial column and the solar radiation ($cos(SZA)$) before 12:20 local time at Xianghe (left) and Maïdo (right), respectively. R is the correlation coefficient, N is the measurement number, and the black dashed line is the linear fitting.

The daily means are calculated from only the measurements between 13:30 and 14:30 local time. The one hour time window around the maximum of the tropospheric NO partial column is used to reduce the impact of the large diurnal variation of the stratospheric NO partial column as we found in Section 3.1. Since the time coverage is relatively short at Xianghe, we assume that there is no long-term trend ($A_1 = 0$).

The mean and std of the stratospheric NO partial columns between 13:30 and 14:30 are $3.6\pm0.5\times10^{13}$ $molecules/cm^2$ at Xianghe, and $4.0\pm0.4\times10^{13}$ $molecules/cm^2$ at Maïdo, respectively. The mean of the stratospheric partial columns at Maïdo is larger than that at Xianghe. The seasonal variation of NO is determined by the equilibrium between NO and $NO_2$ ($NO_x$ ) on the one hand and the reservoir substances, such as $N_2O_5$, $HNO_3$, $ClONO_2$, on the other hand (Jacob, 1999; Vaughan et al., 2006). The FTIR measurements show that the stratospheric NO partial column is high in summer and low in winter, with a peak-to-peak amplitude of $1.1\times10^{15}$ $molecules/cm^2$ at Xianghe and $0.8\times10^{15}$ $molecules/cm^2$ at Maïdo. Keep in mind that the summer at Maïdo is December-February as it is located in the southern hemisphere.

The decrease in the stratospheric NO partial columns between 2013 and 2019 at Maïdo is observed by the FTIR measurements (-0.42 $\pm$ 0.57 %/yr), although the decrease is insignificant as the annual change is within the uncertainty. Galytska et al. (2019) observed a significant decrease in $NO_2$ from the SCIAMACHY satellite measurements in the southern hemisphere between 2002 and 2012, and Dubé et al. (2020) also found a negative $NO_x$ trend in the southern hemisphere derived from SAGE II-OSIRIS satellite measurements and the WACCM model from 2005 to 2014. Although the time coverages of the FTIR measurements at Maïdo and the two previous studies are not the same, all these studies show a consistent negative trend in $NO_x$ in the stratosphere at the latitude of Maïdo.

### 3.3 Comparison with MIPAS measurements

The MIPAS satellite observed the atmospheric NO concentrations globally between 2002 and 2012. There are two spectral resolutions for the MIPAS spectra: 0.05 cm$^{-1}$ before January 2005 named full spectral resolution (FR) mode and 0.121 cm$^{-1}$ after January 2005 named reduced resolution (RR) mode. In this study, we only use the NO MIPAS data after 2005, and the versions are V5r_NO_220 and V5r_NO_221. As MIPAS has a limb view, the MIPAS only provides NO profile above $\sim$ 10 km. The NO profile is retrieved from MIPAS spectra at 5.3 $\mu$m (Bermejo-Pantaleón et al., 2011). The vertical resolution of the NO profile is about 4-6 km, and the uncertainty of the NO profile in the altitude range of 20-60 km is 5-40 % (Sheese et al., 2016).

There are no overlap MIPAS measurements with the FTIR measurements, as Xianghe and Maïdo FTIR both started measuring after 2012. The MIPAS satellite has two windows overpassing one location (around 10:30 and 22:30 local time). The NO stratospheric partial column observed by MIPAS during the night (22:30) is about $1\times10^5$ times less than that observed during the day (10:30). In this section, we select all the MIPAS measurements between 2005 and 2012 within $\pm$ 2° latitude and $\pm$ 2° longitude around each FTIR site and only use the daytime-overpass measurements to compare with FTIR measurements. To take the vertical sensitivity of the FTIR retrieval into account, the MIPAS NO vertical profile is smoothed with the FTIR AVK (Rodgers and Connor, 2003). To reduce the influence from the diurnal variation in the stratospheric NO partial columns

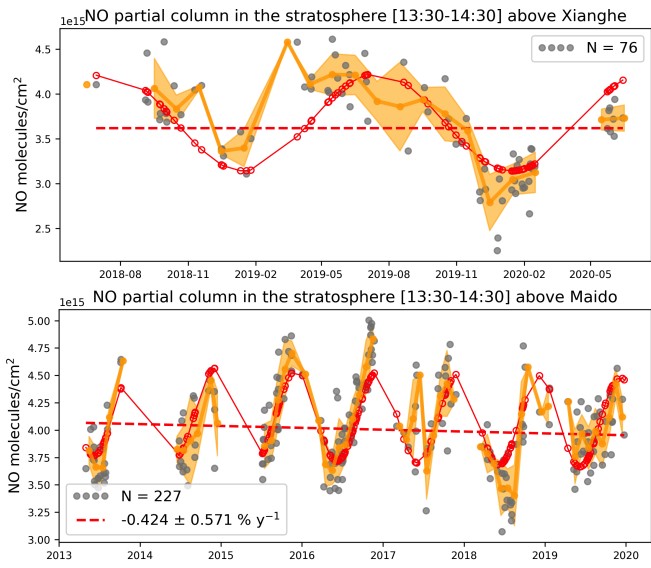

**Figure 8.** The time series of the FTIR NO retrieved stratospheric partial column daily means (grey dots) and monthly means and stds (yellow dots and shadow) at Xianghe (upper) and Maïdo (lower) for only the measurements between 13:30 and 14:30 local time. N is the measurement days. The red dashed line is the $A_0 + A_1 t$ in Eq. (3), the red dots and the red solid line are the fittings with the seasonal variation.

(Figure 6), the FTIR measurements used to compare with MIPAS measurement are limited to the measurements between 9:30 and 11:30.

Figure 9 shows the time series of the stratospheric NO partial columns observed by FTIR and MIPAS measurements above Xianghe and Maïdo. The seasonal variations of the stratospheric NO partial columns observed by MIPAS and FTIR mea-

5 surements are similar, with a high value in summer and a low value in winter. The amplitudes of the seasonal variations of the stratospheric NO partial columns above Xianghe and Maïdo observed by the MIAPS measurements are $1.65 \times 10^{15}$ $molecules/cm^2$ and $1.19 \times 10^{15}$ $molecules/cm^2$, respectively, which are larger than those observed by the FTIR measurements of $1.13 \times 10^{15}$ $molecules/cm^2$ above Xianghe and $1.09 \times 10^{15}$ $molecules/cm^2$ above Maïdo. The possible reason for this difference is that the retrieval uncertainty is relatively large for MIPAS measurements, as we see many large values above

10 both sites. The uncertainties of the MIPAS stratospheric NO partial column are 20.5% and 24.7% above Xianghe and Maïdo, respectively.

The MIPAS measurements show that stratospheric NO was increasing above Xianghe and decreasing above Maïdo between 2005 and 2014, which is consistent with the negative $NO_x$ trends in the southern hemisphere and the positive $NO_x$ trends in the northern hemisphere observed by SAGE II-OSIRIS satellite measurements between 2005 and 2014 (Dubé et al., 2020). Above

15 Xianghe, it is impossible to derive the long-term trend from the FTIR measurements because of the limited measurements. Above Maïdo, the annual relative change of stratospheric NO partial columns observed by MIPAS measurements between 2005 and 2012 is -0.62±0.77 %/yr, which is close to that of -0.68±0.36 %/yr observed by the FTIR measurements between

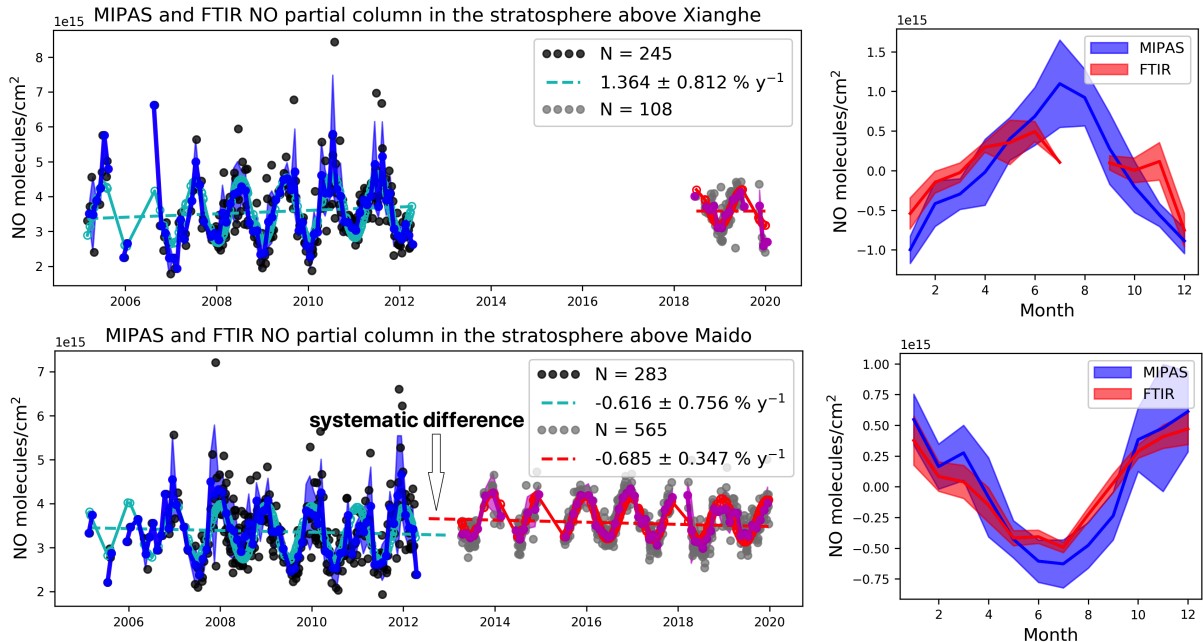

**Figure 9.** Left panels: the time series of the stratospheric NO partial columns observed by MIPAS measurements (black dots: daily means; blue solid line: monthly mean; blue shadow: monthly std; cyan dashed line: trend fitting) and FTIR measurements (grey dots: daily means; purple solid line: monthly mean; purple shadow: monthly std; red dashed line: trend fitting) above Xianghe (upper) and Maïdo (lower). Right panels: the seasonal variations of the stratospheric NO partial columns observed by MIPAS measurements. The MIPAS measurements are selected within $\pm 2°$ latitude and $\pm 2°$ longitude around the sites. Numbers of the measurement days (N) are 245 and 283 from MIPAS at Xianghe and Maïdo, respectively. The FTIR measurements are selected with the measurement time between 9:30 and 11:30, with 108 and 565 days at Xianghe and Maïdo, respectively. The annual changes of NO partial columns derived from MIPAS measurements at Xianghe and Maïdo in 2005-2012, and from the FTIR measurements at Maïdo in 2013-2019. A systematic difference is detected between the MIPAS and FTIR NO stratospheric partial columns.

2013 and 2019. However, we observe a systematic difference of $0.35 \times 10^{15}$ $molecules/cm^2$ between MIPAS and FTIR measurements, which corresponds to 10.6% relative to MIPAS data and 9.6% relative to FTIR measurements. The difference is within the uncertainties of both the MIPAS and FTIR measurements.

## 4 Tropospheric NO partial column

In this section, we only use the tropospheric NO partial columns retrieved by the FTIR measurements with a DOF larger than 0.5 in the troposphere at Xianghe. Figure 10 shows the scatter plots between the tropospheric NO partial columns and stratospheric NO partial columns. When the DOF is larger than 0.5 in the troposphere, the R between retrieved tropospheric NO partial columns and stratospheric NO partial columns is only 0.11, indicating that there is no linear relationship between

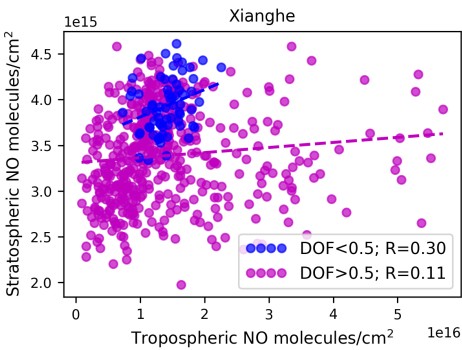

**Figure 10.** Scatter plots at Xianghe between the tropospheric NO partial columns and stratospheric NO partial columns with the DOF less than 0.5 (blue) and greater than 0.5 (magenta) in the troposphere. R is the correlation coefficient. Only the results at Xianghe is presented here as there is no retrieval with a DOF larger than 0.5 in the troposphere at Maïdo.

the retrieved tropospheric and stratospheric partial columns, and the retrieved tropospheric and stratospheric partial columns are almost independent. When the DOF is less than 0.5 in the troposphere, the R between retrieved tropospheric NO partial columns and stratospheric NO partial columns is slightly larger (R=0.30). The reason for this increased correlation is that the retrieved tropospheric partial column is sensitive to the stratosphere, and the individual averaging kernels become broader.

Figure 11 shows the time series of the tropospheric NO partial columns at Xianghe. There is no tropospheric NO measurement in summer due to the high water vapor columns (Figure 5). In addition, due to the COVID-19 lockdown, FTIR NO measurements are not available between 17 February and 23 May 2020. The mean (std) of the tropospheric NO partial columns are $1.4\pm1.0\times10^{16}\ molecules/cm^2$ at Xianghe. The low NO partial column is close to 0, and the high value can reach up to $5.8\times10^{16}\ molecules/cm^2$. There is no clear diurnal variation of the tropospheric NO partial columns derived from the FTIR

measurements, but there is a large day-to-day variation of NO tropospheric partial columns, especially in winter. The NO partial column during November 2019 - February 2020 is generally lower than that in the previous year. It is found that the FTIR tropospheric NO partial column during the COVID-19 lockdown is much less than that before the lockdown. Moreover, the mean FTIR tropospheric NO partial column between 24 January and 16 February in 2020 is $6.1\times10^{15}\ molecules/cm^2$, which is less than the tropospheric NO partial column of $1.1\times10^{16}\ molecules/cm^2$ during the same period in 2019. The decrease in

tropospheric NO partial column during the COVID-19 lockdown period observed by FTIR measurements at Xianghe is generally consistent with the 25-33% decrease in $NO_2$ column observed by TROPOMI and OMI satellite measurements (Bauwens et al., 2020), and the 28-48% decrease in $NO_2$ surface concentration derived from the air pollution sites in Beijing (Wang et al., 2020).

As Xianghe is located in a polluted area, many species have the same anthropogenic sources. For example, both NO and

carbon monoxide (CO) are emitted by combustion from manufacturing and road transportation (Crippa et al., 2018). The FTIR observed spectra at Xianghe are used to retrieve CO following the NDACC-IRWG recommended method (Zhou et al., 2019). The DOF of the retrieved CO profile is about 2.2, and there is independent information in the tropospheric CO partial column

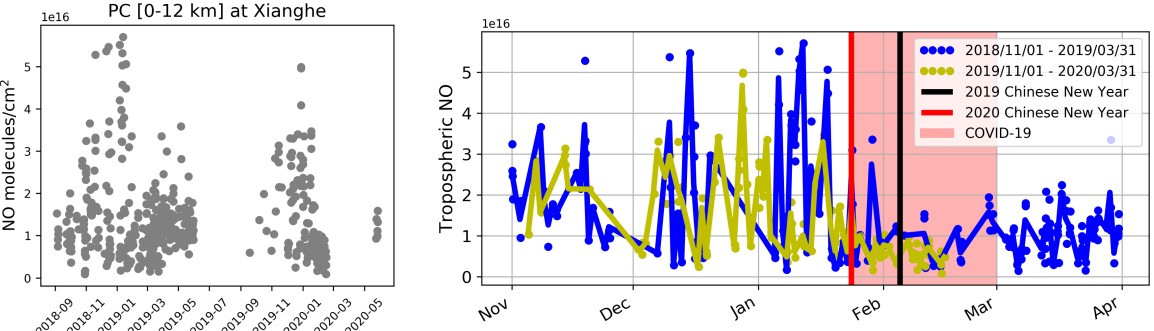

**Figure 11.** Left: the time series of the FTIR retrieved tropospheric NO partial columns at Xianghe. Right: the time series of FTIR retrieved tropospheric NO partial columns during November 2018 - March 2019 and November 2019 - March 2020. The blue and yellow dots are individual measurements and the solid lines are daily means.

(Zhou et al., 2018). Figure 12 shows the correlation between the daily means of FTIR retrieved CO and NO partial columns in the troposphere in winter at Xianghe. The large tropospheric CO and NO partial columns are observed simultaneously. The R is 0.70, indicating that the FTIR measurements can capture the tropospheric NO partial column variability on a synoptic scale.

Although NO and CO have common emission sources, they are very different species in terms of lifetime, chemistry, and transport. Therefore, we also compare the FTIR NO measurements with the ground-based Multi-Axis Differential Optical Absorption Spectroscopy (MAX-DOAS) $NO_2$ measurements in the troposphere at Xianghe. The MAX-DOAS instrument is operated at the same building as the FTIR instrument at Xianghe. The MAX-DOAS measurements can provide the lower tropospheric $NO_2$ partial columns (0-4 km). For more information about the MAX-DOAS $NO_2$ retrieval technique, we refer to Hendrick et al. (2014) and the references therein. Due to an instrument fail, there is no data between August 2018 and September 2019 for the MAX-DOAS measurements. Nevertheless, we collect all the co-located FTIR NO and MAX-DOAS $NO_2$ measurements. Figure 12 shows the correlation between the daily means of FTIR retrieved NO and MAX-DOAS retrieved $NO_2$ partial columns in the lower troposphere (0-4 km) at Xianghe, with the R of 0.86. A good agreement between the NO and $NO_2$ is observed, and it confirms that the FTIR retrieved tropospheric partial columns are reliable.

## 5 Conclusions

In this study, the ground-based FTIR solar spectra at Xianghe and Maïdo are applied to retrieve NO using the SFIT4 algorithm, with a focus on the NO partial columns in the troposphere and stratosphere. Xianghe is a polluted site with a high NO mole fraction near the surface, while Maïdo is a background site with a very low NO mole fraction near the surface. The systematic and random uncertainties of the retrieved NO total column are estimated as 10.3% and 13.5% at Xianghe, and 10.3% and 4.2% at Maïdo. The DOF of the retrieved NO profile is 2.3±0.4 at Xianghe and 1.3±0.1 at Maïdo. The systematic and random

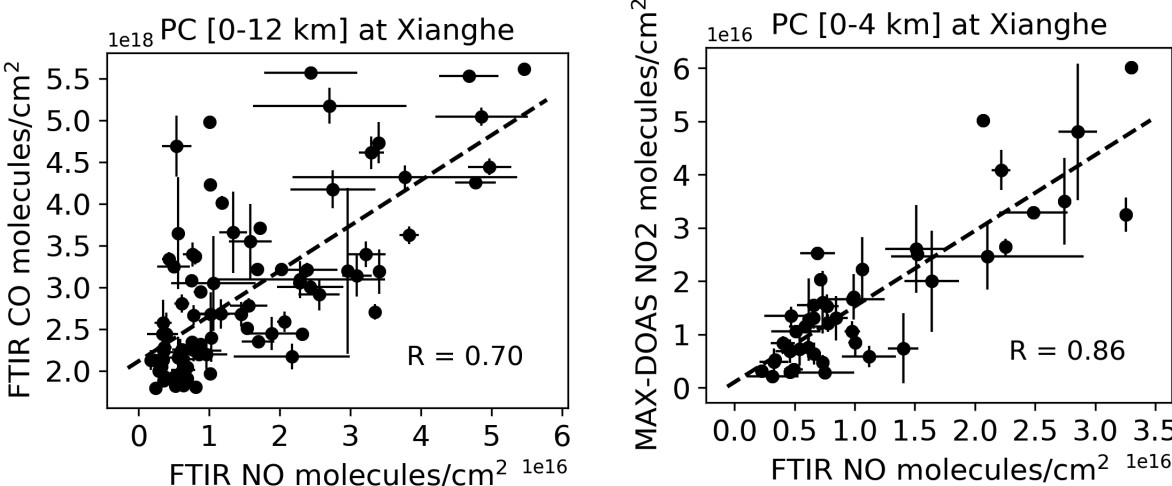

**Figure 12.** Left: the correlation between the daily means of the FTIR retrieved NO tropospheric partial columns and the retrieved CO tropospheric partial columns in winter at Xianghe. Right: the correlation between the daily means of the FTIR retrieved NO partial columns and MAX-DOAS retrieved NO$_2$ partial columns between the surface and 4 km at Xianghe. The error bar is the daily std, and the black dashed line is the linear fit.

uncertainties of the retrieved NO partial columns at Xianghe are estimated as 10.5% and 18.0% in the troposphere, and 10.2% and 4.4% in the stratosphere.

At both sites, we can obtain the NO partial column in the stratosphere from the FTIR retrievals. The FTIR retrievals are able to derive the diurnal variation of the NO partial column in the stratosphere during the daytime, especially at Maïdo. It is found

that the stratospheric NO partial column increases with time in the morning to about 12:20 and there is a linear relationship between the stratospheric NO partial column and the solar radiation intensity, with the R of 0.80 at Xianghe and 0.74 at Maïdo. The stratospheric NO partial column starts decreasing after about 14:00 at Maïdo, but at Xianghe it is hard to observe such consistent change of the stratospheric NO partial column after 14:00 due to the limited measurements. As there is a large diurnal variation in the stratospheric NO partial column, we use the measurements between 13:30 and 14:30 to derive the

seasonal cycle of the stratospheric NO partial column. It is found that the phases of the seasonal variations of the stratospheric NO partial column at these two sites are similar with a high value in local summer and a low value in local winter. Moreover, the FTIR NO partial columns in the stratosphere are compared with the MIPAS satellite observations. After taking the diurnal variation of NO into account, the stratospheric NO partial columns from co-located FTIR and MIPAS measurements show similar seasonal variations at both sites. Above Maïdo, the decrease rate observed by the MIPAS measurements between 2005

and 2012 is close to that observed by the FTIR measurements between 2013 and 2019. The systematic difference between the MIPAS and FTIR measurements is about 10%, which is within their uncertainties.

The tropospheric NO partial column can be retrieved at the polluted site (Xianghe) but not at the background site (Maïdo). We select the retrieval with a DOF in the troposphere larger than 0.5 to calculate the tropospheric NO partial column. Since

the SNR of the spectrum is highly dependent on the $H_2O$ abundance, the successfully retrieved tropospheric NO is generally under a dry condition with a $H_2O$ total column less than $5.7 \times 10^{22}\ molecules/cm^2$ at Xianghe. As a result, the tropospheric NO partial column is very difficult to retrieve in summer. The mean and std of the tropospheric NO partial columns at Xianghe are $1.4 \pm 1.0 \times 10^{16}\ molecules/cm^2$. The mean FTIR tropospheric NO partial column during the COVID-19 lockdown in 2020

is lower than that before the lockdown period, and also lower than that during the same period in 2019. Large tropospheric $NO_2$ (or CO) and NO partial columns are observed simultaneously, indicating that the synoptic variation in the tropospheric NO partial columns can be well captured from the FTIR retrievals at Xianghe. It is the first study of a successful analysis of NO in the troposphere from a ground-based FTIR site. The tropospheric and stratospheric NO retrieval might be possible at other potential FTIR sites inside/near large cities with enhanced levels of NO near the surface.

*Data availability.*  The MIPAS data publicly available https://www.imk-asf.kit.edu/english/308.php. The FTIR NO retrievals at Xianghe and Maïdo are available upon request to the authors.

*Competing interests.*  The authors declare that they have no conflict of interest.

*Acknowledgements.*  This FTIR measurements at Xianghe is funded by the National Natural Science Foundation of China (41975035). The FTIR site at Reunion Island are operated by the BIRA-IASB and locally supported by LACy/UMR8105, Université de La Réunion. The

authors would like to thank Rebecca Buchholz (NCAR) for providing the CAM-Chem model data, and Francois Hendrick (BIRA-IASB) for providing the MAX-DOAS $NO_2$ measurements. EM is a Senior Research Associate with the F.R.S. - FNRS (Brussels, Belgium).

*Author contributions.*  MZ wrote the manuscript. MZ, MDM designed the experiment. MZ, BD and EM discussed the conceptualization. CH, NK, JMM and PW collected the FTIR measurements at Xianghe and Maïdo. BL and CV investigated about the FTIR retrieval strategy. All the authors read and commented on the manuscript.

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
