# Peer review of "Tropospheric and stratospheric NO retrieved from ground-based FTIR measurements"

_Atmospheric Measurement Techniques, 2021_

## Author Comment (AC1)

**Black: referee's comments green: authors' answers**

**First of all, we want to thank the referee 1 for the detailed analysis of our paper. For the details, please look into the paper with keeping track of changes.**

The manuscript "Tropospheric and stratospheric NO retrieved from ground-based FTIR measurements" by Zhou et al presents findings in the retrieval of NO using ground-based solar absorption FTIR at two sites, Xianghe (polluted) and Maido (background/pristine), contrasting the retrieval sensitivity. The work presented fits well within the scope of this journal. Below I have a short list of comments/suggestions that the authors may want to consider for the final version.

**Major comments**

1. I find that the manuscript lacks sufficiently novel findings in the retrieval strategy. As authors mentioned in the introduction, until now, there are few studies focusing on FTIR NO retrieval and past studies have shown little sensitivity in the troposphere. However, in this work authors show results of a single retrieval strategy, i.e., a single micro-window of NO2 has been adapted from past studies, e.g., Notholt et al. (1995). Since the manuscript tries to show the retrieval of NO in the troposphere I was expecting to see a thorough description of optimized windows and retrieval parameters, instead authors only mentioned what was included but do not show an optimization. Are there more microwindows appropriate for NO?. If there are no other suitable windows for NO I highly suggest mentioning it. In general, a description of the optimization is missing.

Thanks for the suggestion. Before finalizing the retrieval window, we have looked at all the NO absorption lines (Figure A1). The NO lines at 1900 cm-1 is selected because of the strongest line intensity and they are less affected by H2O. The observed FTIR spectra (Figure 2 in the AMTD paper) show that the line intensity on the left side of 1900 cm-1 (1850 - 1900) is less than that on the right side of 1900 cm-1 (1900 - 1950), because of the H2O absorption and the optical filter (NDACC Filter 0). As a result, we focus on the windows in the spectral range between 1900 and 1950 cm-1. In fact, only the spectra around 1900 and 1930 cm-1 are less affected by H2O. We tested several windows around 1930 cm-1, and compared the retrievals with that from 1900 cm-1 window. It is found that the uncertainties of the retrievals from 1930 cm-1 region are much larger than that of the retrieval from 1900 cm-1. Therefore, in the end, we choose the 2 strongest NO lines around 1900 cm-1 as the retrieval window, which are the same as the previous studies (Notholt et al., 1995 and Wiacek et al. 2006).

We have added more information in the revised version.

"The strong NO absorption lines are between 1800 and 1950 cm-1. In order to select strong NO lines and to reduce the interference from  $H_2O$ , several windows have been tested. We find that the NO absorption lines at 1900 cm-1 are the optimal choice for ground-based FTIR NO retrieval at Xianghe and Maido, which have been used in the previous studies (Notholt et al., 1995 and Wiacek et al. 2006)."

Figure A1. The intensities of the NO absorption lines from the HITRAN2016 linelist.

2. It is concluded that tropospheric NO is not well retrieved during the summer because of high water vapor abundance. My understanding from the manuscript is that high water vapor reduces the signal to noise ratio. From my previous comment, is there a region where water vapor has less influence?

**Unfortunately, the answer is No.**

Also, I highly recommend checking the zenith angle dependency in the summer vs winter. In my opinion, it may have an effect, for example when the zenith angle is high the optical path may be more sensitive to lower tropospheric air mass, hence NO may be retrievable during high zenith angles. Do measurements in winter vs summer cover the same zenith angle ranges?

Agree that the SZA is also important for FTIR NO retrieval. But the dominate factor caused the significant difference in summer and winter is the water vapor abundance.

With similar SNR of spectra, the DOF of the NO profile increases with SZA. The reason for the increased DOF is that the optical path becomes large with a high SZA, so that more information of NO is retrieved. However, in summer, the SNR of the spectrum is very low due to a high H2O column. Figure A2 shows that only a few successful NO retrievals are available in JJA (summer) at Xianghe. The converged retrievals in summer are generally with a small SZA ( $<50^{\circ}$ ) and a low H2O column. Non-converged retrievals (failed) are generally with a large SZA in summer, because the slant column of H2O increases with the air mass factor ( $\sim 1/cos(SZA)$ ). In winter, the H2O column is much lower, and we can get the converged NO retrieval even with a large SZA ( $>50^{\circ}$ ). In summary, the dominant difference between two spectra in summer and winter with a similar SZA, such as 60°, is the SNR, which is affected by the H2O column difference between summer and winter at Xianghe. Figure 2 in the AMTD paper shows that the spectra at 1900 cm-1 are almost saturated with a high H2O column. As a result, in a wet condition, we are not able to retrieve NO for both tropospheric and stratospheric parts.

Figure A2. The FTIR NO retrievals from all the spectra in JJA (above the dashed line) and DJF (below the dashed line) at Xianghe, with the converged retrievals colored with yellow and the failed retrievals colored with grey. The H2O total columns in summer are much larger than those in winter. The spectra are recorded with a wider range of SZA in summer as compared to winter.

3. There is a contrast between Xianghe (polluted) and Maido (background), however it is not mentioned what would be the detection limit of the NO using these observations. Please include an assessment in the detection limit.

Thanks for the suggestion. We agree that it is very useful to give detection limits for tropospheric NO retrievals. However the situation is not so straightforward. This study shows that the NO retrieval depends on the SNR of the spectra and the NO concentration. Assuming that all the spectra are recorded under clear-sky condition, the SNRs are then strongly affected by  $H_2O$  column and SZA. We cannot give an absolute value of NO concentration as the detection limit. Instead, we focus on the retrievals at Xianghe and Maido, and add the discussion about the NO variations at the two sites.

"In summary, we cannot retrieve NO in the troposphere at Maïdo, because the NO mole fraction near the surface (NOsurf) is low, with a typical value of less than 0.1 ppb. At Xianghe, the spectra recorded under a wet condition (mainly occur in summer) do not allow us to retrieve the tropospheric NO either. In winter, all the retrievals at Xianghe provide both tropospheric and stratospheric NO partial columns (Figures 4 and 5). The retrieved NOsurf in winter varies from 1.3 to 47.2 ppb, with a mean of 11.4 ppb and an std of 10.7 ppb. For all the 240 retrievals in winter, the mean of the H2O total column is  $2.3 \times 10^{22}$  molecules/cm2, and the mean of the SZA is 65.3°. A relatively lower NOsurf at Xianghe can be detected under the condition of a low H2O total column and a large SZA. For example, if we select the retrievals with the NOsurf less than 3 ppb (26 out of 240), the mean of the H2O total column becomes  $1.7 \times 10^{22}$  molecules/ cm2, and the mean of the SZA is 68.1°. 4. Authors show correlation between NO and CO measured by the same instruments. While NO and CO may have the same common emission sources ther are very different species, e.g., CO lifetime is significantly larger and can be transported from other regions, etc. Are there any co-located or close-by in-situ measurements of NO that can be used to see tropospheric columns and enhancements?. I would expect some correlation between insitu and retrieved lower tropospheric NO since the averaging kernels show high sensitivity in the boundary layer.

Thanks for the suggestion, unfortunately, there is no NO in situ measurement available at Xianghe. Instead, we add the comparison between the FTIR NO and MAX-DOAS NO2 measurements. An BIRA-IASB/IAP MAX-DOAS instrument is operated at the same building of the FTIR instrument at Xianghe, which observes several air pollutants, including NO2. Figure A3 shows that the colocated FTIR NO and MAXDOAS NO2 partial columns in the lower troposphere (0-4 km) show a good correlation, with the R of 0.86.

---

## Author Response (AR1)

*Black: referee's comments* *green: authors' answers*
*First of all, we want to thank the referee 1 for the detailed analysis of our paper.*
*For the details, please look into the paper with keeping track of changes.*

The manuscript "Tropospheric and stratospheric NO retrieved from ground-based FTIR measurements" by Zhou et al presents findings in the retrieval of NO using ground-based solar absorption FTIR at two sites, Xianghe (polluted) and Maido (background/pristine), contrasting the retrieval sensitivity. The work presented fits well within the scope of this journal. Below I have a short list of comments/suggestions that the authors may want to consider for the final version.

**Major comments**

1.  I find that the manuscript lacks sufficiently novel findings in the retrieval strategy. As authors mentioned in the introduction, until now, there are few studies focusing on FTIR NO retrieval and past studies have shown little sensitivity in the troposphere. However, in this work authors show results of a single retrieval strategy, i.e, a single micro-window of NO2 has been adapted from past studies, e.g., Notholt et al. (1995). Since the manuscript tries to show the retrieval of NO in the troposphere I was expecting to see a thorough description of optimized windows and retrieval parameters, instead authors only mentioned what was included but do not show an optimization. Are there more micro-windows appropriate for NO?. If there are no other suitable windows for NO I highly suggest mentioning it. In general, a description of the optimization is missing.

Thanks for the suggestion. Before finalizing the retrieval window, we have looked at all the NO absorption lines (Figure A1). The NO lines at 1900 cm$^{-1}$ is selected because of the strongest line intensity and they are less affected by $H_2O$. The observed FTIR spectra (Figure 2 in the AMTD paper) show that the line intensity on the left side of 1900 cm$^{-1}$ (1850 - 1900) is less than that on the right side of 1900 cm$^{-1}$ (1900 - 1950), because of the $H_2O$ absorption and the optical filter (NDACC Filter 0). As a result, we focus on the windows in the spectral range between 1900 and 1950 cm$^{-1}$. In fact, only the spectra around 1900 and 1930 cm$^{-1}$ are less affected by $H_2O$. We tested several windows around 1930 cm$^{-1}$, and compared the retrievals with that from 1900 cm$^{-1}$ window. It is found that the uncertainties of the retrievals from 1930 cm$^{-1}$ region are much larger than that of the retrieval from 1900 cm$^{-1}$ , because the line intensities around 1930 cm$^{-1}$ are 5-10 times less that at 1900 cm$^{-1}$.  Therefore, in the end, we choose the 2 strongest NO lines around 1900 cm$^{-1}$ as the retrieval window, which are the same as the previous studies (Notholt et al., 1995 and Wiacek et al. 2006).

We have added more information in the revised version.

 "The strong NO absorption lines are between 1800 and 1950 cm$^{-1}$. In order to select strong NO lines and to reduce the interference from $H_2O$, several windows have been tested. We find that the NO absorption lines at 1900 cm$^{-1}$ are the optimal choice for ground-based FTIR NO retrieval at Xianghe and Maido, which have been used in the previous studies (Notholt et al., 1995 and Wiacek et al. 2006)."

[Figure]

Figure A1. The intensities of the NO absorption lines from the HITRAN2016 linelist.

2. It is concluded that tropospheric NO is not well retrieved during the summer because of high water vapor abundance. My understanding from the manuscript is that high water vapor reduces the signal to noise ratio. From my previous comment, is there a region where water vapor has less influence?.

Unfortunately, the answer is No.

Also, I highly recommend checking the zenith angle dependency in the summer vs winter. In my opinion, it may have an effect, for example when the zenith angle is high the optical path may be more sensitive to lower tropospheric air mass, hence NO may be retrievable during high zenith angles. Do measurements in winter vs summer cover the same zenith angle ranges?

Agree that the SZA is also important for FTIR NO retrieval. But the dominate factor caused the significant difference in summer and winter is the water vapor abundance.

With similar SNR of spectra, the DOF of the NO profile increases with SZA. The reason for the increased DOF is that the optical path becomes large with a high SZA, so that more information of NO is retrieved. However, in summer, the SNR of the spectrum is very low due to a high $H_2O$ column. Figure A2 shows that only a few successful NO retrievals are available in JJA (summer) at Xianghe. The converged retrievals in summer are generally with a small SZA (<50°) and a low $H_2O$ column. Non-converged retrievals (failed) are generally with a large SZA in summer, because the slant column of $H_2O$ increases with the air mass factor (~1/cos(SZA)). In winter, the $H_2O$ column is much lower, and we can get the converged NO retrieval even with a large SZA (>50°). In summary, the dominant difference between two spectra in summer and winter with a similar SZA, such as 60°, is the SNR, which is affected by the $H_2O$ column. Therefore, we highlight that the dominant factor affecting the FTIR NO retrieval is the $H_2O$ column difference between summer and winter at Xianghe. Figure 2 in the AMTD paper shows that the spectra at 1900 cm$^{-1}$ are almost saturated with a high $H_2O$ column. As a result, in a wet condition, we are not able to retrieve NO for both tropospheric and stratospheric parts.

[Figure]

Figure A2. The FTIR NO retrievals from all the spectra in JJA (above the dashed line) and DJF (below the dashed line) at Xianghe, with the converged retrievals colored with yellow and the failed retrievals colored with grey. The $H_2O$ total columns in summer are much larger than those in winter. The spectra are recorded with a wider range of SZA in summer as compared to winter.

3. There is a contrast between Xianghe (polluted) and Maido (background), however it is not mentioned what would be the detection limit of the NO using these observations. Please include an assessment in the detection limit.

Thanks for the suggestion. We agree that it is very useful to give detection limits for tropospheric NO retrievals. However the situation is not so straightforward. This study shows that the NO retrieval depends on the SNR of the spectra and the NO concentration. Assuming that all the spectra are recorded under clear-sky condition, the SNRs are then strongly affected by $H_2O$ column and SZA. We cannot give an absolute value of NO concentration as the detection limit. Instead, we focus on the retrievals at Xianghe and Maido, and add the discussion about the NO variations at the two sites.

"In summary, we cannot retrieve NO in the troposphere at Maïdo, because the NO mole fraction near the surface ($NO_{surf}$) is low, with a typical value of less than 0.1 ppb. At Xianghe, the spectra recorded under a wet condition (mainly occur in summer) do not allow us to retrieve the tropospheric NO either. In winter, all the retrievals at Xianghe provide both tropospheric and stratospheric NO partial columns (Figures 4 and 5). The retrieved $NO_{surf}$ in winter varies from 1.3 to 47.2 ppb, with a mean of 11.4 ppb and an std of 10.7 ppb. For all the 240 retrievals in winter, the mean of the $H_2O$ total column is $2.3 \times 10^{22}$ molecules/cm$^2$, and the mean of the SZA is 65.3°. A relatively lower $NO_{surf}$ at Xianghe can be detected under the condition of a low $H_2O$ total column and a large SZA. For example, if we select the retrievals with the $NO_{surf}$ less than 3 ppb (26 out of 240), the mean of the $H_2O$ total column becomes $1.7 \times 10^{22}$ molecules/ cm$^2$, and the mean of the SZA is 68.1°.

4. Authors show correlation between NO and CO measured by the same instruments. While NO and CO may have the same common emission sources ther are very different species, e.g., CO lifetime is significantly larger and can be transported from other regions, etc. Are there any co-located or close-by in-situ measurements of NO that can be used to see tropospheric columns and enhancements?. I would expect some correlation between in-situ and retrieved lower tropospheric NO since the averaging kernels show high sensitivity in the boundary layer.

Thanks for the suggestion, unfortunately, there is no NO in situ measurement available at Xianghe. Instead, we add the comparison between the FTIR NO and MAX-DOAS $NO_2$ measurements. An BIRA-IASB/IAP MAX-DOAS instrument is operated at the same building of the FTIR instrument at Xianghe, which observes several air pollutants, including $NO_2$. Figure A3 shows that the co-located FTIR NO and MAXDOAS $NO_2$ partial columns in the lower troposphere (0-4 km) show a good correlation, with the R of 0.86.

[Figure]

Figure A3. The correlation between the daily means of the FTIR retrieved NO tropospheric partial columns and the MAX-DOAS retrieved $NO_2$ tropospheric partial columns. The error bar is the daily std, and the black dashed line is the linear fit.

**Specific comments**

In the abstract: Nitric oxide (NO) is a key active trace gas in the atmosphere, which contributes to form "bad" ozone (O3) in the troposphere and to the destruction of "good" O3 in the stratosphere. I highly recommend avoiding good/bad ozone. Instead, something like this:

"Nitric oxide (NO) is a key active trace gas in the atmosphere, which contributes to form harmful ozone (O3) in the troposphere and to the destruction of O3 in the stratosphere"

Done

Sometimes ozone is spelled other times O3 is used, be consistent in the manuscript and I highly recommend using "ozone".

Done.

P2,l26. It is mentioned that at Xianghe the NO is high, please include values.

Added.

P2, l31. Is there a reason for only using MIPAS? Why not compare it to ACE-FTS?

Because of the occultation geometry, the overpass time of ACE-FTS is about 06:00 and 18:00 at local time. There are no FTIR measurements at these two overpass windows. In order to reduce the large diurnal variation of the stratospheric NO partial column (Figure 4), we compare FTIR and MIPAS measurements. The overpass time of MIPAS is about 10:30 and 22:30, and we use the daytime measurements to compare with the co-located FTIR measurements.

P3. In the description of FTIR sites, please add additional information such as resolution of FTIR measurements, time resolution, i.e., how often do you measure in the region of interest?. Are there additional in-situ measurements of NO around Xianghe & Maido?. It is mentioned that NOx annual emission at Xianghe is one of the largest around the world, could you include typical concentrations comparing Xianghe and Maido?

More information is added now.

P4, Figure 1. The obs-cal is shown in the upper plot but in the bottom plot they are not shown, maybe adding the obs and calc in the bottom plot would be more clear.

We prefer to keep it unchanged. Adding the obs and calc spectra makes the plot very busy.

P4. I have several comments/suggestions regarding the NO a priori profile:

- CAM-Chem is used at Xianghe because WACCM underestimates surface NO concentration. However, CAM-Chem is used up to 50 km, which potentially makes the stratospheric a priori different from WACCM. I wonder why CAMChem was not used only in the lower troposphere then WACCM to use similar a priori for Maido and Xainghe?. Did you assess the impact of different a priori profiles in the stratosphere?

The difference between the CAM-Chem and WACCM is mainly in the troposphere. In the stratosphere, the difference between CAM-Chem and WACCM is relatively small (within 10%). We have tested both CAM-Chem and WACCM as the a priori profile in the stratosphere at Xianghe, the relative difference of NO total column is less than 0.5%.

- Likely NO shows a strong seasonal cycle, did you assess monthly prior profiles?

We prefer to use the fixed a priori. Figure 3 shows that the FTIR retrieval can well capture the NO changes even with a low a prior profile in the stratosphere. In addition, the NO is not only changing with the season, but also with the local hour. The fixed a priori profile can help us to reduce the impact of the a priori information when looking at the seasonal and diurnal variations of NO.

P5, L2-L8. How is the SNR defined?. Are the spectra compared in the summer/winter taken at a similar zenith angle?, maybe I miss it but how does the SNR affect the DOFs?

"The SNR is defined as the ratio of the maximum intensity of the spectra in the NO retrieval window to the root mean square error of the spectra in the noise window between 1650 and 1700 cm$^{-1}$". – added.

The discussion about the SZA see above (the responds to the major comment 2).

"The covariance matrix of the measurement is calculated as $1/SNR^2$ for the diagonal values and 0 for the off-diagonal values. As a result, the retrieval information is strongly affected by the SNR. " – added.

P6, l10. It is mentioned that summer retrievals are limited. It is concluded that the decrease of DOFs in the summer is due to low SNR because of greater water vapor columns. One more thing to assess, in my opinion, is the dependency in the sza, could you please indicate if measurements over summer & winter cover similar zenith angles?. Maybe the optical path using high zenith angles has more sensitivity over the lower troposphere, hence greater DOFs?. In general, the sza dependency is missing and may also contribute to low DOFs in the summer.

See the reply to the major comment 2.

P9, l12-15. First it is described that NO decreases after 14:00 for some months, e.g. January but different for other months, e.g., February. Please add a reason for this. Furthermore, if the fittings are not robust I suggest removing them.

The fittings at Xianghe are not robust due to the lack of measurements, especially before 9:00 and after 16:00. Following the suggestion, and the fittings at Xianghe are removed now.

P13, l4. I suggest adding, maybe next to Figure 9, the monthly mean seasonal variation of MIPAS and FTIR (since there are no coincident dates). This would allow the reader to see the difference in amplitudes mentioned in the text.

Done

P15. L2-9. In the context of Figure 10, the manuscript indicates that when the DOF is larger than 0.5 in the troposphere there is no linear relationship between the retrieved tropospheric and stratospheric partial column. However, I do see it as enhancements of NO2 in the troposphere are not correlated with stratospheric NO2, how do you disentangle the atmospheric chemistry and the retrieval DOFs?

We suppose that the referee is talking about NO instead of $NO_2$.

Due to different physical and chemical progresses, there is no direct link between the tropospheric and stratospheric NO. Therefore, if the FTIR retrieval can separate the NO in the troposphere and stratosphere, it is expected to observe a weak correlation between the tropospheric and stratospheric partial columns (this is the case for the retrievals with DOF larger than 0.5 in the troposphere). Otherwise, if the FTIR retrieval is not able to separate the NO in the troposphere and stratosphere, the retrieved NO in the troposphere is then affected by the NO signal in the stratosphere so that there is a relative large correlation between the tropospheric and stratospheric partial columns (this is the case for the retrievals with DOF less than 0.5 in the troposphere).

Have you explored Figure 10 but color coded by SZA?

As discussed above, the SZA is not the dominate parameter here.

*Black: referee's comments green: authors' answers*
*First of all, we want to thank the referee 2 for the detailed analysis of our paper.*
*For the details, please look into the paper with keeping track of changes.*

**General Comment:**

The paper by Zhou et al is a report of column measurements of NO from two sites, one in a polluted area of the Northern Hemisphere (Xianghe, China), and the other in a remote part of the Southern Hemisphere (Maido, ReUnion Island). These data also represent a polluted urban area (in the troposphere) and non-polluted site. This would appear to be the first report of a successful analysis of NO in the troposphere from a ground based FTIR, a nuiance that the authors do not explicitly state. Ground-based NO columns has been reported before in the literature, and invariably from NDACC sites that do not in general, see the sort of heightened levels of NO that is reported at Xianghe. So while the stratospheric columns and comparison with satellite data is not new, the tropospheric partial columns of NO are unique, at least as a first report in the literature. Similar data at other Chinese sites may exist and indeed, other potential NDACC sites near large cities that may or may not have enhanced levels of NO near the ground, but the potential is there to explore.

Thanks for the suggestion. We address the importance of this study in the revised version by adding the following sentence:

"We present the first study of a successful analysis of NO in the troposphere from a ground-based FTIR site. The tropospheric and stratospheric NO retrieval might be possible at other potential FTIR sites inside/near large cities with enhanced levels of NO near the surface."

The methods used follow reasonably standard software procedures that have been developed over many years within the NDACC, but NO is not one of the normal target molecules reported by this network. The authors here represent an experienced team who have a very good track record in this area of atmospheric spectroscopy. The paper is not claiming to provide an extensive description of their method, but refer to a few papers in the literature where this is done. A few more details on how they derived some of the parameters used in the analysis, should be fleshed out a bit, as is mentioned below in the comments sections.

Given that NO2 is an integral part of the NOx family along with NO, it would have been an obvious addition to add NO2 to this analysis. This added molecule is readily available in the FTIR spectra, as the authors know, so this would have been an obvious choice to make alongside CO. Or alternatively, in a city like Xianghe, are there air quality monitors like a NOx box that measures NO/NO2?

FTIR $NO_2$ retrievals at Maido and Xianghe are still under investigation within the EU-ACTRIS framework. Therefore, we did not discuss the FTIR $NO_2$ retrieval here. We are still working on the FTIR $NO_2$ harmonization within the NDACC-IRWG network, including Maido and Xianghe, and will present the result in a separate study.

There are no such air quality monitors measuring NO and $NO_2$ simultaneously. But, there is a nearby BIRA-IASB/IAP MAX-DOAS at Xianghe, providing $NO_2$ columns. The MAX-DOAS tropospheric $NO_2$ measurements at Xianghe have been used for satellite validation and atmospheric pollution studies (Hendrick et al., 2014;Verhoelst et al., 2021). In the revised

version, the co-located MAX-DOAS $NO_2$ measurements are compared with FTIR NO measurements in the lower troposphere at Xianghe. A good agreement between the FTIR NO and MAX-DOAS $NO_2$ partial columns in the vertical range between 0 and 4 km, with the R value of 0.86. The high correlation with $NO_2$ is encouraging.

There is also the question of why there is not a modelling component to this paper? So really the question is: is this paper about a new measurement capability (tropospheric NO), or a comparison between a polluted and non-polluted site, or a satellite comparison, or what? So before this paper is published, the purpose of this paper and the new novel aspects need to be clearly pointed out.

Thanks for the suggestion. To make the target of the study clear, the following state has been added in the introduction.

"The aims of this study are 1) to investigate whether it is possible to retrieve NO partial columns in the troposphere and stratosphere from the ground-based FTIR measurements, especially at the polluted site Xianghe; 2) to better understand the diurnal, synoptic and/or seasonal variations of NO partial columns in the stratosphere (and troposphere if possible) observed by the ground-based FTIR measurements at Xianghe and Maido, together with other measurements, such as co-located satellite measurements."

The level of written English in general ok, but there are a few grammatical issues which are listed in the comments.

Thanks a lot for correcting the grammatical issues.

**Specific comments:**

1. P1, L7: " …almost not able to be retrieved …" => "…is very difficult to retrieve…"

Done

2. P1, L20: "basically" => "mainly"

Done

3. P2, L2: "The stratospheric…" => "Stratospheric…"

Done

4. P2, L4: "…(Park et al,. 2012), the stratosphere…"  =>  "…(Park et al,. 2012), stratosphere…"

Done

5. P2, L23: "…even so for …" => "… even for…"

Done

6.  P3, L4: "…to Beijing." => "…of Beijing"

Done

7.  P3, L6: "…recording the near …" => "… recording near …"

Done

8.  P4, L11: define WACCM with a reference.

Done

9.  P4, L16: place this definition and reference to WACCM in line 11.

This is the definition of CAM-Chem not WACCM.

10. P4, L18: "…above that is still taken…" = > "…above 50 km is taken …"

Done

11. P4, L19: expand a bit on the Tikhonov equation. It is entered here without explaining any of the terms. Explain how a value of 50 was obtained.

Done

12. P5, L7: "The HBR cell …" => "HBr cell …"

Done

13. P5, L15: …"several less…" => "…several orders of magnitude less…"

Done.

14. P5, L16: this sentence would read better as; "Therefore, in the stratosphere the FTIR retrievals during the daytime are much larger than the a priori profile."

Done.

15. P5, L19: "…have the sensitivity…" => "…have sensitivity…

Done

16. P5, L20: This is a little misleading the way this is written. Not all layers are sensitive to the stratosphere, since there is no information in the troposphere. A more correct way to put this is that there is sensitivity to NO in the layers in the stratosphere. Note also some sensitivity in the upper troposphere between 10 and 16km, particularly at Maido.

Agree, the sentence is reworded now.

17. P6, L2: Presume this is the average dofs over the entire datasets?

Yes, "over the entire datasets" is added now.

18. P7, L9: "...to the HITRAN2016..." => "...to the HITRAN2016 linelist..."

Done

19. P8, fig 4 caption: "...DOF equalling.." => "...DOF's equal ..."

Change to "DOF of 0.5"

20. P8, L4: "...estimated 13.5%..." => "...estimated to be 3.5%..."

Done

21. P8, L8: "...less than that of NO..." => " ...less than the NO..."

Change to "The random uncertainty of NO stratospheric partial column is less than the random uncertainty of NO total column"

22. P8, L9: " ...less ..." => "smaller"

Done

23. P9, L5: suggest this sentence reads "Due to photochemical reactions (Kondo et 5 al., 1990), a large diurnal variation of the stratospheric NO is expected."

Accepted.

24. P9, L7: "...SZA of measurements." => "...SZA of the measurements."

Done

25. P9, L7: "...2 order..." => "... 2nd." There are a few other locations where this appears.

Done

26. P9, L8: "...t is in a fraction of local hour)." => "...t is a fraction of the local hour)."

Done

27. P9, L12: "...with the time." => "...with time."

Done

28. P9, L18: "...formed NO..." => "... NO formed..."

Done

29. P9, L20: "...stratosphere, then ..." => "....stratosphere, so..."

Done

30. P9, L23: "...and of 0.74..." => "...and 0.74..."

Done

31. P10, error budget: what about inferring species? A solar model is used (but not mentioned as part of the retrieval strategy, for example table 1) so presumably this is part of the retrieved parameters. But does this solar model include both solar line strength and shift?

Added now.

32. P11, fig 7 caption: "The R is the ..." => "R is the ..."

Done

33. P12, L1: "...t is in fraction of year..." => "...t is fraction of the year..."

Done

34. P12, L2: "...which is relative..." => "...is relative..."

Done

35. P12, L10: "...on one hand..." => "...on the one hand..."

Done

36. P12, L29: "...both start measuring..." => "...both started measuring..."

Done

37. P12, L34: "...smoothed with FTIR..." => "...smoothed with the FTIR..."

Done

38. P13, L5: "...are similar observed..." => "...are similar as observed..."

Changed to "The seasonal variations of the stratospheric NO partial columns observed by MIPAS and FTIR measurements are similar, with a high value in summer and a low value in winter"

39. P13, L9: "...The possible reason is that..." => "...The possible reason for this difference is that..."

Done

40. P14, figure 9: The key needs to be reasonably self-explanatory. The numbers and trends in the key should be in the figure caption. For example the black dot entry should read MIPAS daily means, and the number of points can go into the figure caption. Same comment for all the other entries. The colour coding is also not consistent between what is described in the caption and what appears on the graph. For example, the blue shadow for MIPAS is actually purple, the blue solid line for MIPAS is green, while the purple shadow for the FTIR is pink. This could be related to the way colours are displayed in the pdf reader.

Done. Color is corrected, and More information is added in the figure caption.

41. P14, L2: "…which is corresponding…" => "…which corresponds…"

Done

42. P15, figure 10 caption: "Scatter plots between…" => "Scatter plots at Xianghe between…"

Done

43. P15, figure 10 caption: a comment about the way this plot is presented and captioned. The explicit way of knowing that this figure represents Xianghe is the caption title, which is fine But the caption explanation should be more explicit about what the data is and where is from since there is more than one site.

Thanks for the comments. More information is added in the caption.

44. P15, L8: "…slightly large…" => "…slightly larger…"

Done

45. P15, L8: "It is because that the…" => "The reason for this increased correlation is …"

Done

46. P15, L7-9: The underlying reason is the increased cross-relation between the tropospheric and stratosphere layers, due to the individual averaging kernels being broader.

Right, it is also added now.

47. P15, L11: why are there no tropospheric NO measurements in summer? This maybe explained later (high water?), but a reference could be placed here that this will be explained later in the paper.

Done.

48. P15, L12: would this normally be expressed as mean (std) is 1.4 (1.0) x 10^16, as it is in the abstract.

Done

49. P16, figure 11 caption, last sentence: this colour is not yellow, more light green. Maybe this is a function of the pdf viewer?

The colors are fine on my PC.

50. P16, figure 12 caption: "…CO tropospheric partial columns." => "CO tropospheric partial columns at Xianghe."

Done

51. P16, L2: "…combustion for…" => "…combustion from …"

Done

52. P16, L4: individual => independent

Done

53. P17, L25: depend => dependent

Done

[revised manuscript text omitted]